# Antimicrobial Peptides: A Potent Alternative to Antibiotics

**DOI:** 10.3390/antibiotics10091095

**Published:** 2021-09-10

**Authors:** Mariam Rima, Mohamad Rima, Ziad Fajloun, Jean-Marc Sabatier, Burkhard Bechinger, Thierry Naas

**Affiliations:** 1Team ReSIST, INSERM U1184, School of Medicine Université Paris-Saclay, 94270 Le Kremlin-Bicetre, France; mariamrima6@gmail.com; 2Laboratory of Applied Biotechnology, Azm Center for Research in Biotechnology and Its Applications, EDST, Lebanese University, Tripoli 1300, Lebanon; mohamad.rima@hotmail.com (M.R.); ziad.fajloun@ul.edu.lb (Z.F.); 3Department of Biology, Faculty of Sciences III, Lebanese University, Tripoli 1300, Lebanon; 4Institut de Neuro Physiopathologie, UMR7051, Aix-Marseille Université, Faculté de Pharmacie, 27 Boulevard Jean Moulin, 13005 Marseille, France; 5Institut de Chimie de Strasbourg, CNRS, UMR7177, University of Strasbourg, 67008 Strasbourg, France; bechinge@unistra.fr; 6Institut Universitaire de France (IUF), 75005 Paris, France; 7Bacteriology-Hygiene Unit, Assistance Publique/Hôpitaux de Paris, Bicêtre Hospital, 94270 Le Kremlin-Bicetre, France; 8French National Reference Centre for Antibiotic Resistance: Carbapenemase-Producing Enterobacterales, 94270 Le Kremlin-Bicetre, France

**Keywords:** antimicrobial peptides, antibiotic resistance, multidrug resistance, ESKAPE

## Abstract

Antimicrobial peptides constitute one of the most promising alternatives to antibiotics since they could be used to treat bacterial infections, especially those caused by multidrug-resistant pathogens. Many antimicrobial peptides, with various activity spectra and mechanisms of actions, have been described. This review focuses on their use against ESKAPE bacteria, especially in biofilm treatments, their synergistic activity, and their application as prophylactic agents. Limitations and challenges restricting therapeutic applications are highlighted, and solutions for each challenge are evaluated to analyze whether antimicrobial peptides could replace antibiotics in the near future.

## 1. Introduction

There was collective enthusiasm about the advent of antibiotic therapy, and this optimism prevailed in the light of the constant discoveries of new classes of antibiotics, despite the very early description of therapeutic failures and resistance to treatments with these drugs [1]. At the present, in 2021, multidrug-resistant (MDR) and even pan-drug-resistant (PDR) bacteria have spread widely around the world and are currently responsible for increasing morbidity and mortality rates, as well as the significant cost to society [2]. The low frequency of discoveries of new classes of antibiotics, and the rapid emergence of resistance to novel antibiotics, show the need for novel therapeutic alternatives to antibiotics, such as lysine-based small molecules, vaccines, anti-virulence strategies, phage therapy and antimicrobial peptides.

Antimicrobial resistance (AMR) is an emerging global health problem that results, in some cases, in difficulties to treat bacterial infections. It was listed by the World Health Organization (WHO) among the top ten global public health threats facing humanity, as it is predicted to cause about 10 million deaths each year by 2050 [3]. Therefore, efforts to slow down the propagation of AMR have been implemented worldwide. As such, a Global Action Plan on Antimicrobial Resistance (GAP) was created in 2015, aiming to implement national action plans to limit the progress of AMR. In addition, the WHO reports call for urgent action to avert an antimicrobial resistance crisis and insists on the importance of discovering and developing new antibiotics. Thus, new compounds that are active against pathogens, especially those which cause nosocomial infections and tend to adopt multidrug resistance, are needed. To curb this problem, several alternative therapies have been proposed, among which antimicrobial peptides (AMPs) were suggested to be very promising more than 20 years ago, as they have existed in nature for millions of years with almost no or limited resistance development [4]. This makes them very attractive compared to antibiotics that develop resistance relatively fast. This absence/slow development of resistance against microbes may be attributed to the presence of various modes of action of AMPs against the bacteria in comparison to the fixed targets used by the antibiotics [5]. In addition, AMPs are considered less toxic, as they are broken down into amino acids unlike other therapeutics, which might generate potentially harmful metabolites. This review is to highlight where these molecules stand now in the overall scheme to curb MDR bacterial infections. Their potential to counteract AMR, to replace traditional antibiotics, to evaluate their benefits and to describe the challenges faced by R&D will be discussed in this review.

AMPs are small polypeptide molecules, typically made up of around 12 to 50 amino acids, found in all classes of living organisms [6]. These molecules are produced as secondary metabolites, are part of the innate immunity and are, in mammals, usually ribosomally produced by epithelial cells, but also by phagocytes (cells of the immune system). These peptides can be found in tissues or mucous membranes; in fact, the latter harbor a multitude of pathogenic or commensal microorganisms. Among these peptides, some have a broad spectrum of antimicrobial activity, capable of inhibiting or killing different types of microorganisms (MO) (Gram-positive or -negative bacteria and/or fungi) but also protozoans, and viruses [7,8]. According to The Antimicrobial Peptide Database, more than 3257 antimicrobial peptides have been described to date [9]. They originate from six kingdoms, where 365 come from bacteria, 5 from archaea, 8 from protists, 22 from fungi, 360 from plants, and 2414 from animals, in addition to some synthetic peptides. Recently, another webserver with several subsections that screen among 40,000 entries has been made available. It summarizes all available data for desired peptide properties, thus allowing further structural and functional studies [10].

## 2. Classification and Mode of Action of Antimicrobial Peptides

### 2.1. Classification

AMPs are usually classified according to several criteria: First, based on their biological source. In this classification, we distinguish AMPs from human and mammalian sources, such as cathelicidin and defensin, which make up the most important families of AMPs [11]; AMPs from amphibians [12], fish [13], insects [14], and plants [15] are also classified in this category. Classification could be based on the AMP’s biological functions; for example, antibacterial, antiviral, antifungal, antiparasitic peptides were described. Third, based on their biochemical properties (amino acid sequence, composition, length, hydrophobicity, charge), where conformation and structure serve as criteria for classification as well [7].

### 2.2. Mode of Action

Some factors can modulate the activity and specificity of AMPs, such as size, charge, hydrophobicity, secondary structure, or amphiphilic character. The conformation of AMPs may also play a role in antimicrobial activity. Indeed, it has been shown that peptides possessing amphipathic structures interact better with the membrane of pathogens. AMPs can have a membrane permeabilization action and/or act on certain intracellular functions (Figure 1) [16].

#### 2.2.1. Membrane Permeabilization

The bacterial membrane is the most important target of AMPs as they act by disrupting the integrity of the pathogen’s membrane [17]. Several models have been described in the literature, all of which result in osmotic lysis. Membrane disruption may happen through different models, as recently reviewed in Huan et al. [7]. (i) The “barrel-stave” model indicates that peptides insert into the membrane and direct their hydrophobic regions toward the lipid core of the bilayer, forming a transmembrane pore. Experimental evidence for this model has been obtained from structural and biophysical investigations of very hydrophobic sequences, some being devoid of any charge [18]. (ii) The “carpet” model suggests that AMPs accumulate parallel to the membrane surface, forming a “carpet” [19]. Once a threshold concentration of peptides has been reached, they exert a “detergent” effect, leading to the rupture of the cell membrane. Extensive experimental evidence for this mechanism has been obtained for many cationic amphipathic peptides such as magainins [20]. (iii) The “toroidal pore” model implies that AMPs insert themselves perpendicularly into the membrane through interactions between the lipid bilayer and the hydrophilic region of the peptides. In doing so, the membrane distorts and thus forms a “toroidal pore” [21]. (iv) The “aggregate” model assumes the formation of aggregates of peptides and lipids, allowing the translocation of AMPs across membranes. When considering these models, it should be kept in mind that on the one hand, lipid bilayers are soft and can adapt their shapes and thicknesses to the membrane-inserted peptides. On the other hand, the peptides cover a highly dynamic and flexible conformational space and upon interaction with the membrane also respond. Thereby, many different supramolecular arrangements can be formed depending on the lipid composition, peptide concentration, salt, and buffer. These considerations have been incorporated into the SMART model, which takes into consideration that peptides and lipids adjust their conformation and shape when interacting with each other thereby covering a full range of possibilities of interactions between AMPs and membranes [22].

#### 2.2.2. Inhibition of Intracellular Functions

Some peptides cross the cytoplasmic membrane and target cellular processes essential for the survival of the pathogen, including inhibition of DNA replication, protein synthesis, interference with nucleic acid biosynthesis, metabolism, cell division, cell wall, and LPS binding proteins [23].

Indeed, some cationic AMPs have been shown to complex and flocculate nucleic acids or other anionic macromolecules due to their affinity for negatively charged phosphodiester bonds [24,25]. For example, Frenatin 2.3S internalizes within bacterial cells after destabilizing their membranes and can bind nucleic acids [26]. Others are able, even at low concentrations, to affect protein synthesis. The examples are countless; for instance, tryptophan-containing AMPs were proven to be efficient at killing *Pseudomonas aeruginosa* by down-regulating the expression of DNA replication-initiating genes [27]. In addition, Tur1A, an AMP found in dolphins, binds to ribosomes, and blocks the translation of mRNA into proteins [28]. Thanatin, an insect-derived AMP targets the bacterial LPS and induces the LPS-mediated aggregation [29]. Tridecaptin blocks ATP synthesis in bacteria and is active on MDR and colistin-resistant Enterobacterales [30].

#### 2.2.3. Immunomodulatory Activity

Well characterized for their antimicrobial activities, AMPs are also known for their immunoregulatory functions. They can thus participate in the recruitment and activation of immune cells.

It has been described in the literature that certain AMPs, such as defensins, can enhance the production of inflammatory cytokines, such as interleukin-1 [31]. To mention another example, cathelicidin BF has been shown to exhibit immunomodulatory activity that, in mice, is able to ameliorate pneumonia caused by *Pseudomonas aeruginosa* [32].

#### 2.2.4. Action on Biofilms

When in the form of biofilms, pathogens escape more easily the action of conventional antimicrobials. Indeed, microorganisms in biofilms are capable of tolerating, in particular, *via* transcriptional regulatory mechanisms, high concentrations of antimicrobials even though they are totally sensitive in planktonic conditions [33].

The mode of action of AMPs on biofilms is not yet fully understood; however, it is recognized that they can prevent different stages of biofilm formation. It has been shown that some AMPs can inhibit the adhesion of bacteria to surfaces by reducing certain types of motilities such as “swarming” or “swimming”. They are also able to stimulate “twitching”, another type of motility known to promote the disassembly of biofilms. AMPs can also down-regulate certain genes involved in quorum sensing, the latter being known to play a role in biofilm formation and/or in the organization and communication of bacteria within the biofilm [34]. Thus, it has been demonstrated that LL-37 cathelicidin targets quorum sensing mechanisms that control *P. aeruginosa* biofilm formation [35].

## 3. AMPs as Potential Alternatives to Antibiotics

Several advantages were observed for AMPs over classical antibiotics: (i) usual resistance mechanisms observed toward conventional antibiotics are bypassed by AMPs [36,37]; (ii) they are easier to synthesize since they consist usually of short amino acid sequences [38]; (iii) they show rapid killing [39]; (iv) they act on bacteria irrespective of their resistance phenotype, since they are not affected by the known resistance mechanisms [40,41,42], and (v) they do not affect microbiota, which are often disrupted by conventional antibiotics [43]. These advantages made it possible to have several AMPs that are now in medical use, such as nisin, gramicidin, polymyxins, daptomycin, and melittin (reviewed by Dijksteel et al. [44]).

### 3.1. Activity against MDR Bacteria and ESKAPE

Hundreds of AMPs have shown antibacterial activity against resistant bacteria in vivo [45]. Among the most recent examples, a novel piscidin-like peptide called cerocin from black sea bass fish was described to exert broad-spectrum antimicrobial activity against several bacteria, especially Gram-positive pathogens [46]. Similarly, jelleine-I, a small AMP formed by eight amino acids, acts against both Gram-negative and Gram-positive bacteria, mainly by disrupting the integrity of the cell membrane [47]. Furthermore, another novel defensin-like peptide was described recently for its antimicrobial activity against Gram-positive bacteria, including *Staphylococcus aureus*, *Staphylococcus carnosus*, *Nocardia asteroides*, and one tested Gram-negative bacterium *Psychrobacter faecalis* [48]. Interestingly, AMPs can act against bacteria of the ESKAPE complex, which is made up of six pathogens responsible for most of the nosocomial life-threatening infections: *Enterococcus faecium*, *S. aureus*, *Klebsiella pneumoniae*, *Acinetobacter baumannii*, *P. aeruginosa*, and *Enterobacter* spp. [49]. ESKAPE pathogens are usually MDR, which limits therapeutic options and puts patients’ lives at risk. Many AMPs showed promising activity against ESKAPE pathogens. For example, bip-P-113, dip-P-113 and nal-P-133 are three derivatives of the AMP p-133, that work well against *E. faecium*, showing a low MIC of 4 µg/mL as compared to 64 µg/mL for vancomycin [50].

The resistance of *S. aureus* to methicillin remains a huge drawback associated with high mortality rates [51]. Many AMPs are able to eradicate methicillin-resistant *S. aureus* (MRSA) [52]; for example, poly(2-oxazoline)s, an easy-to-synthesize polymer mimetic of AMPs that displays high and selective antimicrobial activity against MRSA, is considered as a promising antimicrobial candidate due to its low MIC values (12.5 µg/mL) [53]. ΔM3 is a novel synthetic peptide consisting of a short amino acid sequence that has low toxicity coupled to a great biological activity, especially against *S. aureus* strains. The low MICs of ΔM3 (5 µg/mL) against *S. aureus* (ATCC25923) and MRSA (7.5 µg/mL) [54], make it a great AMP candidate. Finally, SAAP-148, a derivative of the human LL37 peptide, is active against several resistant ESKAPE pathogens (*S. aureus*, *P. aeruginosa*, *A. baumannii*), including biofilms that occur in wound infections [55].

*K. pneumoniae*, is a challenging bacterium with respect to conventional antibacterial therapy, as it may be encapsulated, limiting the antibiotic’s capacity to penetrate. However, this bacterium is sensitive to pepW, an AMP that targets, aggregates and disrupts *K. pneumoniae’s* capsules with low MICs (2–4 µg/mL) [56]. PepW is also active against *Escherichia coli* with even lower MICs of 1–2 µg/mL [56]. Similarly, AA139 and SET-M33 may be promising novel drugs against MDR *K. pneumoniae* strains with MICs ranging from 4 to 16 µg/mL, but more interestingly, they remained active even on colistin-resistant *K. pneumoniae* isolates [57].

Aurein 1.2, CAMEL, citropin 1.1, LL-37, Cec4 and omiganan have shown excellent activity against *A. baumannii,* with MICs going from 2 to 16 µg/mL [58,59]. Consequently, AMPs are considered as potential candidates for new treatments since they have relatively low MICs against a bacterial strain for which only limited, if any, treatment options are available. As reviewed by Neshani et al., 46 AMPs exhibit good activity against *A. baumannii*, some of which could be used as preventive and others as therapeutic options [60]. DN4 and DC4 have shown significant activity against *A. baumannii* as well as in a *K. pneumoniae* model, where an MIC of 32 µg/mL was noted [61].

On the other hand, ZY4 permeabilizes the MDR *P. aeruginosa* membrane and exhibits killing with MICs ranging from 2 to 4.5 µg/mL [62]. GL13K was also described as an active AMP showing good activity against MDR *P. aeruginosa*, with an MIC of 8 µg/mL [63,64]. Furthermore, NCK-10 is a small molecule mimetic made of a naphtalene aromatic ring, a decyl chain and a cationic lysine with potent *P. aeruginosa* antibiofilm activity [65].

### 3.2. Synergistic AMPs

To date, and according to the AMPs database, 47 of the listed AMPs show synergistic activities against several pathogens. This synergistic/additive effect is of interest to find new efficient therapeutic strategies against the ESKAPE pathogens [66]. Synergies may be observed between different AMPs, AMPs of the same family or AMPs and antibiotics. Table 1 summarizes a few examples of AMPs that showed good synergistic effects.

### 3.3. AMPs Role in Preventive Healthcare

#### 3.3.1. Prophylaxis

Another potential domain of action of AMPs is preventive healthcare, which deals with the prevention of illness [78]. As such, polycationic peptides were used as prophylactic agents against methicillin-susceptible or methicillin-resistant *Staphylococcus epidermidis* [79]. Antibacterial peptide-based gels were also synthesized to prevent medical implanted device infections [80]. Likewise, coating antimicrobial peptides was used for the prevention of urinary catheter-associated infections [81]. For example, bactericidal activity of chain 201D, an AMP derived from crowberry endophytes, was effective against *E. coli* and *S. aureus*, when immobilized on a model surface where it can bind and kill, by contact, a high percentage of adherent bacteria [81]. This gives AMPs a high potential for the development of antimicrobial surfaces, namely for application in urinary catheters [81]. Another example of AMP-based infection prevention in healthcare is dental caries caused by several pathogens [82]. Some AMPs including α-helical ones prevent bacterial adherence to implant surfaces [83].

#### 3.3.2. Biofilm Treatment

Biofilms are one of the most common spreaders of infectious disease [84]. They form when communities of microorganisms adhere to some surfaces, and they are usually hard to treat [85]. Biofilms can develop on catheters, breast implants, or even on living tissue such as lungs or teeth, putting patients at risk of developing nosocomial infections that are difficult to eradicate with antibiotics [86]. By living in biofilms, bacteria are protected from harmful conditions, and thus can develop resistance to common antibiotics, which makes AMPs promising alternatives for new therapeutic strategies [87,88]. In this context, as already reviewed by Galdiero et al., many AMPs serve as agents that interfere with bacterial biofilm formation/expansion [89], inhibit bacterial adherence on indwelling medical devices, and therefore can reduce nosocomial infections [81,90]. For example, vancomycin-coated tympanostomy tubes resist MRSA biofilm formation [91]. Interestingly, another AMP of the cathelecidins family D-LL-31 greatly enhances the biofilm-eradicating effects of currently used antibiotics [92,93]. Furthermore, SAAP-148, a synthetic AMP, showed great antibiofilm activity, where it eradicates MRSA and MDR *A. baumannii* [55]. Similarly, KKd-11 is a remarkable AMP that was not only able to inhibit the formation of biofilms but also effectively kill the bacteria within the biofilms [94]. Together, these findings show interesting potential of AMPs, which is a step forward as compared to conventional antibiotic therapy.

#### 3.3.3. Vaccination: AMP-Based Vaccines

In the context of the preventive healthcare field, AMPs could also be used as vaccines or vaccine adjuvants to induce protective immunity or immune response against infections [95]. As such, IC31, a synthetic adjuvant containing the AMP KLKL5KLK, can induce a protective immune response against the desired antigen [96]. It was also proven to be a potent adjuvant to a DNA vaccine targeting *Mycobacterium tuberculosis* [95]. However, no vaccine exists until today against MDR bacterial isolates [97]. Therefore, in this present scenario of exponentially increasing AMR, using the immuno-modulatory advantages of AMPs could help in alleviating AMR problem. The development of AMP-based vaccines may be able to reduce the burden and to overcome the AMR issue worldwide [98].

## 4. AMPs for Diagnostic Purposes

Rapid diagnostic tools are the key to prevent the spread of contagious infectious agents by rapidly identifying carriers or infected patients and prompt implementation of adapted infection control measures. The ability of AMPs to sense bacterial presence in potentially infected clinical samples is a major advantage. As already reviewed by Pardoux et al., AMPs could serve as infection selective tracers where they act as probes in biosensors and thus detect pathogens [99]. This is due to the preferential binding of AMPs along with their great number allowing the coverage of a wide range of microorganisms. AMPs were also described as powerful radiotracers and proved to be even better than labeled antibiotics [100]. [99mTc-HYNIC/EDDA]-MccJ25 is an example of an antimicrobial peptide analog that was used as a potential radiotracer for detection of infection [99]. Interestingly, this imaging agent for *E. coli* infection is also stable against enzymatic and metabolic degradation [101].

## 5. AMPs Facing 2019 Pandemic: SARS-CoV-2

In addition to activity toward bacteria, according to the database of antimicrobial activity and structure of peptides (DBAASP) [102]., 208 AMPs have been reported to show antiviral activities against SARS-CoV-2. Slight changes to glycocin F and lactococcin G increased their efficacy against SARS-CoV-2 by inhibiting protein synthesis with no side effects [103]. Similarly, mucroporin-M1 (LFRLIKSLIKRLVSAFK), a peptide analog displaying four amino acid changes (G3R, P6K, G10K, and G11R) as compared to the parent peptide mucroporin shows therapeutic activities against SARS-CoV-2 [104]. Brilacidin and LF are additional drug candidates for SARS-CoV-2 treatment [105,106]. Caerin 1.6 and caerin 1.10, two other AMPs, have a very high potential to interact with the spike surface protein of SARS-CoV-2, but low affinity for the ACE2 receptor. The selectivity of these peptides toward viral proteins makes them potential candidates for SARS-CoV-2 therapy [107].

## 6. Limitations and Challenges of the Therapeutic Use of AMPs

Despite the numerous advantages of AMPs, several problems prevented them from being used as therapeutic agents in human clinics. For instance, one major problem is their limited stability, especially their susceptibility to the degradation by proteases [108,109]. In fact, the instability toward proteases hampers the development of many AMPs [110], by limiting their use for topical applications and/or requiring sequence modifications, to make them less susceptible to degradation [111]. Other problems include high extraction costs [112,113], poor bioavailability [6], short half-lives [114], cytotoxicity, and lack of specificity [115,116]. In addition, AMPs tend to display lower direct antimicrobial activity as compared to antibiotics. In real, regulatory agencies want AMPs to exhibit similar or even stronger activities than antibiotics [37]. Finally, resistance development by the bacteria toward AMPs cannot be excluded, especially if the microorganism is often exposed to them [117]. So far, only a few resistance mechanisms to AMPs have been described, including sequestration by bacterial enzymes and export by efflux pumps [118]. Taken together, these limitations slow down the successful reach of AMPs into clinical phases and limit their use to a small fraction of their potential (Figure 2) [44].

### 6.1. Clinical Trials of AMPs

Although a huge number of AMPs have been described, very few reach the market. For example, ghrelin, the endogenous appetite peptide, possessing a good antimicrobial activity against Gram-negative bacteria is still in phase II clinical trial [119,120]. Surotomycin, is another example of AMP that is currently under investigation in phase III clinical trials; it targets *Clostridium difficile* and treats the associated diarrhea. The examples are countless; many AMPs remain under investigation until their efficacy is confirmed and secondary effects are eliminated. Some AMPs fail to reach the last clinical trials, such as (i) frulimicin B, which was terminated after phase I clinical trial due to unfavorable pharmacokinetics [44] and (ii) murepavadin, which was eliminated after a phase III clinical trial since it causes acute kidney injuries [44].

### 6.2. How to Overcome Some of These Limitations?

Different strategies were explored to overcome the drawbacks described previously. Modifications such as encapsulation of AMPs could be the key to avoiding their degradation by proteases and thus improving stability [121,122]. Several AMPs have been tested before and after encapsulation, and an obvious enhancement was observed: encapsulation of the AMP IK8, with gold nanorods into polyethylene glycol hydrogels protected the AMP from proteolysis and provided a triggered release of the therapeutic molecule [123]. High extraction costs could be limited by synthesizing AMPs in the lab using natural ones as templates where, at the same time, synthetic AMPs are always an enhanced version of the natural ones [124]. For this, genetic programming could be used to save not only money, but also on time and workload, and this could be started in silico through programs that predict the activity of AMPs before even generating them to avoid unnecessary synthetic efforts [125]. To improve bioavailability, increased half-lives of AMPs, and reduced cytotoxicity, structural modifications such as dimerization [126], methylation and more have been successfully tested [127]. Specificity could be introduced by strategies for site specific release [128], such as specifically targeting AMPs (STAMP), which showed high efficiency in delivering the AMP to the target without harming the natural microbiota [129,130]. This strategy aims to implant AMPs into a drug delivery vehicle composed of the AMP itself in addition to a targeting peptide that allows a specific binding to the desired pathogen [43,129].

### 6.3. Synthetic AMPs

For decades, research for AMPs was not always successful. As mentioned previously, some AMPs were not introduced into clinics due to their toxicity/ degradation, while others were not very performant. However, none of these efforts were lost as these AMPs played the role of templates for the development of more efficient agents. For example, the introduction of L-phenylalanine into protonectin enhanced its selective antibacterial activity against several Gram-positive bacteria and reduced MIC in *S. aureus* from 16 to 4 µg/mL [131]. In addition, the cathelicidin-derived antimicrobial peptide PMAP-23 was further modified by amino acid substitutions to obtain PMAP-23RI (for, Leu5-Arg and Thr19-Ile). The variant exhibited a higher stability and improved antibacterial activity when compared with the original PMAP-23, with MICs reduced by half in some cases from a simple modification to the original AMP [132]. Analogs of mastoparan-C were also designed by amino acid substitution and peptide truncation, which reduced cell toxicity and improved the bioactivity of the original AMP [127,133].

Table 2 provides additional information on many AMPs discussed in this paper.

## 7. Conclusions

AMPs constitute promising agents with good antibacterial activity against a large set of pathogens, including Gram-negative and Gram-positive bacteria. Their ability to kill ESKAPE pathogens, especially those being pan-drug-resistant, represents a clinical added value over antibiotics due to the limited treatment options available to eradicate such infections. In this review, examples of AMPs effective against MDR bacteria that have the potential to replace conventional antibiotics have been reported. Their roles in boosting the activity of therapeutic molecules, eradication of biofilms, and preventive healthcare broaden AMPs’ therapeutic potential (Figure 3). However, despite the huge number of AMPs described, and the efforts to optimize them, only a few reached advanced clinical trials. There are several challenges, such as their low stability, high extraction costs, and cytotoxicity, which need to be overcome so AMPs can be used in clinics. Nevertheless, due to increasing research efforts, it is expected that, hopefully, potent alternatives to antibacterial agents with the ability to eradicate untreatable life-threatening infections will be developed soon.

## Figures and Tables

**Figure 1 antibiotics-10-01095-f001:**
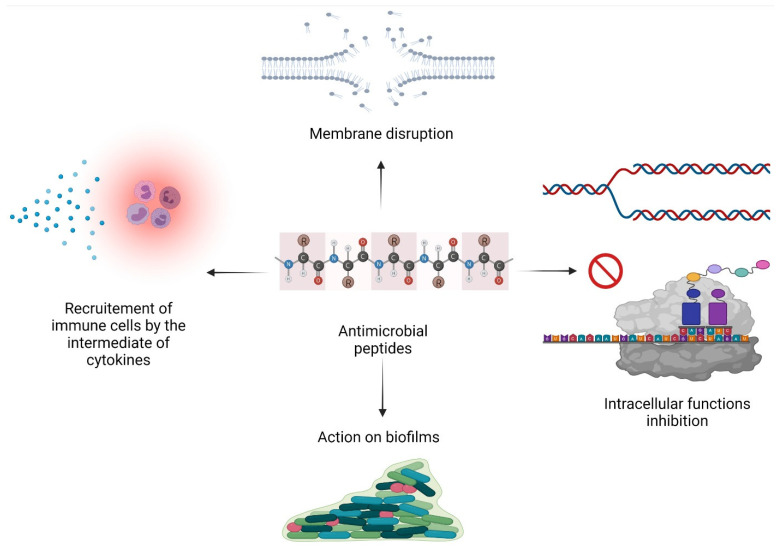
Mechanisms of action of antimicrobial peptides.

**Figure 2 antibiotics-10-01095-f002:**
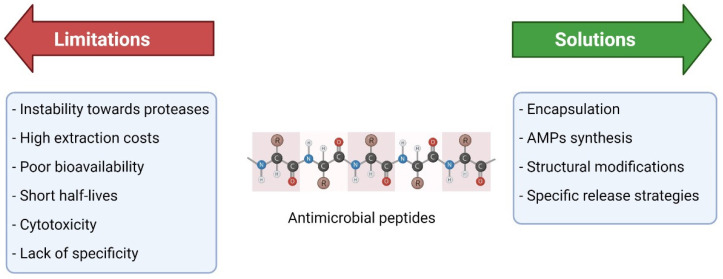
Limitations of AMPs and possible solutions for their therapeutic use.

**Figure 3 antibiotics-10-01095-f003:**
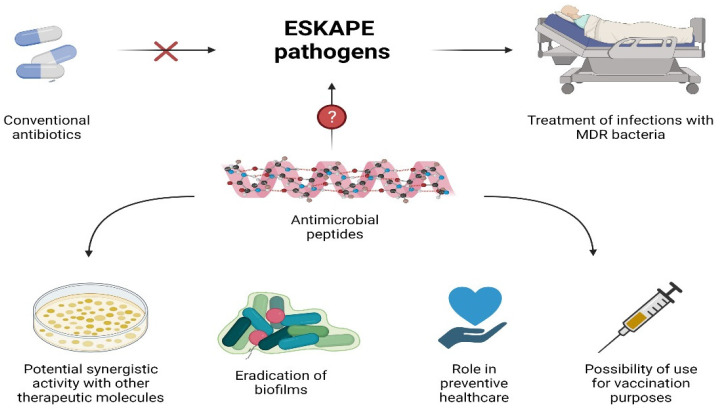
Potential use of AMPs.

**Table 1 antibiotics-10-01095-t001:** Properties of AMPs showing synergistic activities with one another and/or with antibiotics.

AMPs	Source	Synergistic Molecule	Target	Refs.
PGLa	Frog skin	Magainin 2	*E. coli* and *S. aureus*	[67]
Ranalexin	-Bullfrog *R. catesbeiana* *-Staphylococcus simulans*	Endopeptidase lysostaphin	*S. aureus* (MRSA)	[68]
Tridecaptin M	Mud bacterium	Rifampicin, vancomycin, and ceftazidime	Extremely drug-resistant *A. baumannii*	[69]
Dermaseptin	Amphibians skin	Dermaseptin	*E.coli*, *P. aeruginosa*, *S. aureus*	[70,71]
Bactenecin	Lactic acid bacteria	Bactenecin	*E. coli*, *P. aeruginosa*, *S.* Typhimurium	[72,73]
Lactoferricin	Mammalians	Ciprofloxacin, ceftazidime	*P. aeruginosa*	[74]
Nisin	*Lactococcus lactis*	Colistin	*Pseudomonas* biofilms	[75,76]
P10		Ceftazidim/doripenem	MDR *A. baumannii* and colistin-resistant *P. aeruginosa*	[76]
Gad-1	Fish	Kanamycin, ciprofloxacin	*P. aeruginosa*	[77]

**Table 2 antibiotics-10-01095-t002:** Amino acid sequences, type of synthesis and target groups of the AMPs discussed in this paper. Data were retrieved from the database of antimicrobial activity and structure of peptides [102]. GP: Gram-positive bacteria; GN: Gram-negative bacteria.

AMP	Amino Acid Sequence	Synthesis	Target Group
Cathelicidin-BF	KFFRKLKKSVKKRAKEFFKKPRVIGVSIPF	Ribosomal	GP and GN
Defensin	GFGCPLDQMQCHRHCQTITGRSGGYCSGPLKLTCTCYR	Ribosomal	GP and GN
Nisin-P	VXxKXLXxPGXKxGILMXXAIKxAxXGXHFG	Ribosomal	GP and GN
Gramicidin S	VXLfPVXLfP	Nonribosomal	GP, GN, parasites, fungus, cancer cells, and mammalian cells
Polymyxin Colistin	XTXXXlLXXT	Synthetic	GN
Mellitin	GIGAVLKWLPALIKRKRQQ	Synthetic	GP, GN, and mammalian cells
Sm-Piscidin	KGARQAWKDYKYNRNMQKMNQGYGQQGG	Ribosomal	GP, GN, and fungus
Jellein-1	PFKISIHL	Ribosomal	GP, GN, and mammalian cells
Aurein-1.1	GLFDIIKKIAESI	Ribosomal	GP and GN
Citropin-1.1	GLFDVIKKVASVIGGL	Ribosomal	GP, GN, fungus, mammalian cells, and mollicute
Omiganan	ILRWPWWPWRRK	Synthetic	GP, GN, fungus, and mammalian cells
ZY4	VCKRWKKWKRKWKKWCV	Synthetic	GP, GN, fungus, and mammalian cells
GL13K	GKIIKLKASLKLL	Synthetic	GP, GN, insects, and mammalian cells
Chain201D	KWIVWRWRFKR	Synthetic	GP, GN, and fungus
SAAP-148	LKRVWKRVFKLLKRYWRQLKKPVR	Synthetic	GP and GN
Lactococcin-G	GTWDDIGQGIGRVAYWVGKAMGNMSDVNQASRINRKKKHKKWGWLAWVEPAGEFLKGFGKGAIKEGNKDKWKNI	Ribosomal	GP
Caerin-1.10	GLLSVLGSVAKHVLPHVVPVIAEKL	Ribosomal	GP, GN, viruses, and mammalian cells
Murepavadin	LSYXXXXWXXASPP	Synthetic	GP, GN, cancer cells, and mammalian cells
Protonectin	ILGTILPLLKGL	Ribosomal	GP, GN, cancer cells, fungus, and mammalian cells

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
