# Peer review of "Antimicrobial Peptides: A Potent Alternative to Antibiotics"

_antibiotics, 2021, doi:10.3390/antibiotics10091095_

Round 1

Reviewer 1 Report

The manuscript entitled “Antimicrobial Peptides: A potent alternative to antibiotics?” by Rima et al. report recent development in the field of antimicrobial peptides to there potential use as antibiotics. Manuscript contains an introduction followed by the classification, mode of action, and there action as antibiotics against ESKAPE pathogen. AMPs role in preventive healthcare, biofilm treatment, and AMP based vaccines are interesting and well written. However, diagnostic use of AMPs needs more context and some graphical representation. The limitation with AMPs was well explained and provided. Manuscript needs more graphical representation and tables for better representation of text information. Authors should provide a table with AMPs and there target pathogen. Chemical structures of peptides will be helpful to consider. Overall manuscript is well written and will be very helpful to the microbiology and infectious society community. Therefore, the manuscript should be revised for further consideration.

Author Response

Comment 1: Diagnostic use of AMPs needs more context and some graphical representation.

Response: Further explanation was added to the diagnosis paragraph. A figure was added at the end of the review in the conclusion, summing up all the applications of AMPs.

Comment 2: Manuscript needs more graphical representation and tables for better representation of text information

Response: Figure 3 and table 2 were added to the manuscript, they summarize the different potential use of AMPs, and additional information on some AMPs of this reviww, respectively. In addition, a sentence was added to the conclusion to highlight the different applications of AMPs

Comment 3: Authors should provide a table with AMPs and there target pathogen. Chemical structures of peptides will be helpful to consider

Response: A table with the majority of discussed AMPs was included. In this table, amino-acid sequence of each AMP was cited, as well as their target group and their type of synthesis.

Reviewer 2 Report

Dear Authors,

I read your review with high interest. It is very well-written, easy to read and full of useful examples. I believe it is a timely interesting review for non-specialists and those who want to understand the potential and future perspectives of Antimicrobial peptides as an alternative to antibiotics.

I only just a minor comment/suggestion. It would be useful if you add information about the SMART model that is mentioned in line 123. I think such an integrated approach would be very useful if it is better described in this review.

Author Response

Comment 1: It would be useful if you add information about the SMART model that is mentioned in line 123.

Response: A sentence was added to this paragraph to make it clearer and the reference is provided if the reader wants to get more information. The description of different AMP membrane interactions was detailed in the “membrane permeabilization” paragraph. SMART model represents just an approach that unifies all those interactions, and ensures that the properties of AMPs (size, hydrophobicity …) affect the way in which they interact with the membrane.